# Comprehensive Genomic Analysis of *Klebsiella pneumoniae* and Its Temperate N-15-like Phage: From Isolation to Functional Annotation

**DOI:** 10.3390/microorganisms13040908

**Published:** 2025-04-15

**Authors:** Reham Yahya, Aljawharah Albaqami, Amal Alzahrani, Suha M. Althubaiti, Moayad Alhariri, Eisa T. Alrashidi, Nada Alhazmi, Mohammed A. Al-Matary, Najwa Alharbi

**Affiliations:** 1Basic Sciences, College of Science and Health Professions, King Saud Bin Abdulaziz University for Health Sciences, Riyadh 11481, Saudi Arabia; bagmij@ksau-hs.edu.sa (A.A.); alzahraniama@ksau-hs.edu.sa (A.A.); althubaitis@ksau-hs.edu.sa (S.M.A.); alharirimo@mngha.med.sa (M.A.); alrashidiei@kaimrc.edu.sa (E.T.A.); alhazmin@ksau-hs.edu.sa (N.A.); 2King Abduallah International Medical Research Center, Riyadh 11481, Saudi Arabia; 3Ministry of the National Guard-Health Affairs, Riyadh 11426, Saudi Arabia; 4Department of Biological Sciences, Faculty of Science, King Abdulaziz University, Jeddah 21589, Saudi Arabia; malmatry0002@stu.kau.edu.sa (M.A.A.-M.); nmaalharbi@kau.edu.sa (N.A.); 5Department of Animal Production, Faculty of Agriculture, Sana’a University, Sana’a 12191, Yemen

**Keywords:** phage, *Klebsiella pneumoniae*, temperate, functional annotation

## Abstract

Antibiotic resistance to *Klebsiella pneumoniae* poses a major public health threat, particularly in intensive care unit (ICU) settings. The emergence of extensively drug-resistant (XDR) strains complicates treatment options, requiring a deeper understanding of their genetic makeup and potential therapeutic targets. This research delineated an extensively drug-resistant (XDR) *Klebsiella pneumoniae* strain obtained from an ICU patient and telomeric temperate phage derived from hospital effluent. The bacteria showed strong resistance to multiple antibiotics, including penicillin (≥16 μg/mL), ceftriaxone (≥32 μg/mL), and meropenem (≥8 μg/mL), which was caused by SHV-11 beta-lactamase, NDM-1 carbapenemase, and porin mutations (*OmpK37*, *MdtQ*). The strain was categorized as K46 and O2a types and carried virulence genes involved in iron acquisition, adhesion, and immune evasion, as well as plasmids (IncHI1B_1_pNDM-MAR, IncFIB) and eleven prophage regions, reflecting its genetic adaptability and resistance dissemination. The 172,025 bp linear genome and 46.3% GC content of the N-15-like phage showed strong genomic similarities to phages of the Sugarlandvirus genus, especially those that infect *K. pneumoniae*. There were structural proteins (11.8%), DNA replication and repair enzymes (9.3%), and a toxin–antitoxin system (0.4%) encoded by the phage genome. A protelomerase and ParA/B partitioning proteins indicate that the phage is replicating and maintaining itself in a manner similar to the N15 phage, which is renowned for maintaining a linear plasmid prophage throughout lysogeny. Understanding the dynamics of antibiotic resistance and pathogen development requires knowledge of phages like this one, which are known for their temperate nature and their function in altering bacterial virulence and resistance profiles. The regulatory and structural proteins of the phage also provide a model for research into the biology of temperate phages and their effects on microbial communities. The importance of temperate phages in bacterial genomes and their function in the larger framework of microbial ecology and evolution is emphasized in this research.

## 1. Introduction

The development of extensively drug-resistant (XDR) bacterial pathogens poses a major threat to public health worldwide. Now, a major pathogen causing hospital-acquired infections, including urinary tract infections, pneumonia, and bloodstream infections, is *Klebsiella pneumoniae* [1]. The reputation of *K. pneumoniae* as a major nosocomial pathogen has been much boosted by its ability to acquire and disseminate antibiotic resistance genes via horizontal gene transfer routes, including plasmids, transposons, and prophages [2]. Antibiotic resistance genes (ARGs), especially those that code for beta-lactamases and carbapenemases, have made many first-line antibiotics less effective [3]. This makes treatment plans more difficult and puts more stress on healthcare systems [4]. Virulence factors, including siderophores (e.g., yersiniabactin and enterobactin) and adhesion systems (e.g., *E. coli* common pilus), increase the bacterium’s potential to colonize host tissues, avoid immune responses, and start chronic infections [5]. Developing sensible plans to treat *K. pneumoniae* depends on an understanding of the genetic pathways leading to antibiotic resistance and virulence [6]. By means of horizontal gene transfer (HGT) and manipulation of bacterial virulence and resistance traits, bacteriophages, also known as phages, affect bacterial evolution to a large degree [7]. Temperate phages integrate with bacterial genomes as prophages and may contain genes that provide their hosts with selection benefits, such as antibiotic resistance or virulence factors [8]. The linear genome of the N15-like phage distinguishes it from most other temperate phages that do not merge into the host chromosome; it survives as a linear plasmid during the lysogenic stage [9,10]. Protelomerase, which changes the linear phage genome into a circular form during replication, and partitioning proteins, which help the phage genome to be stably segregated during cell division, highlight several mechanisms the phage uses to preserve its genetic material, helping the phage to survive among bacterial populations and increasing their genetic variation [11]. This replication system guarantees consistent phage genome inheritance and helps bacterial hosts horizontally transmit genetic material, including ARGs and virulence factors [11]. Researching these phages helps us grasp the dynamics of host–phage interactions and the processes behind phage-mediated bacterial evolution [8,9]. The phage genome codes for a lot of different proteins, such as structural proteins, DNA replication and repair enzymes, and toxin–antitoxin systems that make it easier for the genome to infect, multiply, and stay inside bacterial hosts [12]. Analyzing temperate phages helps explain bacterial evolution, host–phage interactions, and antibiotic resistance [13]. Phages increase bacterial pathogen genetic diversity and adaptation, which is crucial to understanding antibiotic resistance and pathogen development [8]. The phage’s structural and regulatory proteins mimic temperate phage biology and their effects on microbial communities [14].

Although lytic phages are favored by therapeutics because of their rapid bacterial elimination, temperate phages are promising candidates for therapy and phage engineering because of their ability to integrate into bacterial genomes and carry useful genes [7,15]. Temperate phages may persist in bacteria and disseminate genetic material that can kill bacteria or reduce bacterial virulence or antibiotic resistance [16]. The engineering of therapeutic temperate phages has improved with synthetic biology where temperate phages can acquire CRISPR-Cas technology to delete antibiotic resistance or virulence genes [17,18]. Temperate phages may delete the integration mechanism to become “lytic-only” phages that infect and lyse bacteria [19]. Engineering projects take advantage of the phage linear plasmid (like telomeric phages) in phage therapy and for the expression of recombination proteins [20]. The present investigation examines an N15-like phage identified in hospital wastewater and an extensively drug-resistant *K. pneumoniae* strain obtained from a clinical environment. The genetic and functional traits of the bacterial strain and the phage investigated in this study provide valuable insights into the interactions between bacteria and phages, as well as their implications for pathogen evolution and antibiotic resistance. This study underscores the significance of temperate phages in bacterial genomes and their contributions to microbial ecology and evolution, thereby indicating the necessity for ongoing research into phage–bacteria interactions to combat antibiotic-resistant diseases.

## 2. Material and Methods

### 2.1. Culture Conditions and Bacterial Strains

Intensive care units (ICUs) extensively drug-resistant (XDR) clinical strain of *K. pneumoniae* was isolated from sputum sample. Bacterial identification and the antibiotic susceptibility profiles and MICs of the isolates were assessed by the VITEK 2 system (VITEK^®^ 2, BIOMÉRIEUX, Craponne, France). Bacterial strains were cultivated in Luria–Bertani (LB) broth or on LB agar plates at 37 °C and preserved at −80 °C in glycerol stocks for long-term storage.

### 2.2. Bacteriophage Isolation and Purification

Environmental samples (hospital effluents) were collected in a sterile 1 L sterile glass bottle for phage isolation. The enrichment method was used, where 10 mL of sample was mixed with 10 mL of LB broth containing a bacterial host, incubated at 37 °C for 24 h with shaking, and filtered through a 0.22 µm membrane [21]. The resulting phage suspension was serially diluted and plated on soft LB agar containing the host bacteria to detect plaque formation. Single plaques were picked, purified by repeated plaque assays, and stored at 4 °C.

### 2.3. Phage Characterization

Phage morphology was observed using transmission electron microscopy (TEM) according to standard procedures [22]. Briefly, a drop of the purified phage suspension was placed on a carbon-coated copper grid, stained with 2% uranyl acetate, and examined using a JEOL JEM-1400 electron microscope (Tokyo, Japan) at 120 kV.

The host range of the isolated phage was determined by performing plaque assays against a panel of clinical *K. pneumoniae* strains (from our lab stock) and *E. coli* ATCC 25922™, *K. pneumoniae* ATCC13883™ and *P. aeruginosa* ATCC 9027™. The strains were cultured overnight and mixed with soft agar containing the host bacterial strains, and plaque formation was observed to determine the host range of the phage [23].

### 2.4. Bacterial DNA Isolation, Sequencing and Bioinformatics Analysis

The bacterial genomic DNA of *K. pneumoniae* ST147 Kpn-R1 was isolated using the Qiagen Blood and Tissue DNA Extraction Kit, following the manufacturer’s protocol. DNA purity and concentration were verified using a nanodrop spectrophotometer (Thermo Fisher Scientific, Waltham, MA, USA). Whole-genome sequencing (WGS) was performed by Macrogen, Seoul, Republic of Korea on the Illumina NovaSeq 6000 platform (2 × 150 bp paired-end reads) (Illumina, San Diego, CA, USA). Libraries were prepared using the TruSeq Nano DNA Kit (350 bp insert size) (Illumina, San Diego, CA, USA), with library quality assessed via the Agilent 2100 Bioanalyzer (Agilent Technologies, Santa Clara, CA, USA) and TapeStation D1000 Screen Tape (Agilent Technologies, Santa Clara, CA, USA). Raw sequencing reads were trimmed with Trimmomatic (v0.39) to remove adapters and low-quality bases (parameters: ILLUMINACLIP:TruSeq3-PE-2.fa:2:30:10, LEADING:20, TRAILING:20, SLIDINGWINDOW:4:20, MINLEN:50) [24]. The cleaned reads were assembled into contigs using SPAdes (v3.15.5) with default settings [25]. Raw data from whole-genome sequencing (WGS) were integrated using Proksee Assemble (v1.3.0) [26]. Genome annotation was performed using Prokka (v1.2.0), which identified open reading frames (ORFs), tRNA, rRNA, and other genomic features [27]. Map Builder (v2.0.5) generated a CGView JSON file for showing the completed genome from GenBank formats. To rank BLAST tracks based on similarity and color BLAST features based on percent identity, the BLAST Formatter (v1.0.3) was used, assisting in the identification of conserved and divergent regions.

Alien Hunter (v1.1.0) was used to forecast likely horizontal gene transfer (HGT) events based on atypical nucleotide compositions that indicate alien DNA acquisition [28]. To compare the genome with closely related species and determine taxonomic connections, the whole-genome average nucleotide identity (ANI) was calculated using FastANI (v1.1.0) [29]. The Comprehensive Antibiotic Resistance Database (CARD) Resistance Gene Identifier (RGI) (v1.2.1) was used to identify antibiotic resistance genes. Based on the CARD database, this tool anticipates resistance genes and mutations, giving information on the genome’s antibiotic resistance profile [30]. VirSorter (v1.1.1) was used to find viral genomes [31]. Phigaro (v1.0.1) has been annotated in prophage regions [32]. Mobile OG-db (v1.1.3), a database and tool designed for the identification of mobile genetic elements (MGEs), including plasmids, transposons, and integrons [33]. The CRISPR/Cas Finder (v1.1.0) tool was used to locate CRISPR arrays and associated Cas proteins, offering insights into the genome’s adaptive immune system [34]. The Proksee tool displayed and organized all genetic traits. All visible tracks on the genome map were documented in a detailed figure caption created using the Track List Caption (v1.2.0) [26].

Using Kaptive Web [35], capsule (K) and lipopolysaccharide (O) serotypes in *K. pneumoniae* genomes were predicted. Sequence-based typing and known *Klebsiella* strain comparisons were made using the PubMLST *Klebsiella* database [36]. Virulence factors were identified using the 2019 Virulence Factor Database (VFDB) [37]. Barrnap helped pull out the 16S rRNA gene sequence [38]. By use of NCBI BLASTn database comparison, closely similar species were identified. A phylogenetic tree based on the 16S rRNA sequence and other conserved markers was built using MEGA 12 [39], therefore exposing the evolutionary links of the isolate, with the final genome deposited in GenBank under BioProject PRJNA1217456.

### 2.5. Phage DNA Isolation, Sequencing, and Bioinformatics Analysis

Phage DNA was extracted from purified lysates using the Norgen Biotek Phage DNA Isolation Kit (Norgen Biotek Corp., Thorold, ON, Canada). Sequencing libraries were prepared identically to bacterial methods (TruSeq Nano DNA Kit, Illumina, San Diego, CA, USA; Illumina NovaSeq 6000, Illumina, San Diego, CA, USA). Raw reads were processed with Trimmomatic (same parameters as above) and assembled using SPAdes (v3.15.5). Additionally, the quality of the genome assembly was checked using FastQC (v0.12.1) and QUAST (v5.2.0) [40]. Genome completeness was ascertained using CheckV (v1.0.1) [41], and structural features were projected using PhageTerm (v4.0.0) [42]. Functional annotations included phage lifestyle prediction using Bacphlip (v0.9.6) [43] and protein classification using PhANNs [44]. Comparative genomics was run using Pyani (v0.2.12) [45] and Clinker (v0.0.24) [46]. AMRFinderPlus (v3.12.8) [47] and VirulenceFinder (v2.0.4) [48] separately verified antimicrobial resistance and virulence genes. The phage genome was deposited in GenBank under accession numbers PQ800144.1–PQ800159.1.

### 2.6. Statistical Analysis

Statistical analyses were conducted using R (v4.3.1) and Python (v3.10) to evaluate genomic data. Descriptive statistics summarized key metrics (e.g., genome size, GC content, gene coverage). Comparative genomic analyses included average nucleotide identity (ANI) calculations using FastANI and MASH distances for phylogenetic comparisons. Multiple testing corrections (Benjamini–Hochberg method, FDR < 0.05) were applied where applicable. For phylogenetic analyses, branch support was assessed via 500 bootstrap replicates in MEGA12. Correlation analyses between genetic features (e.g., plasmid replicons, prophage regions) and antibiotic resistance profiles used logistic regression models (R stats package, v4.4.3, R Core Team, 2025). MASH distances were used to show the genome in CGView (v1.1.1) [49] against the Millardlab Phage Database [50]. Phylogenetic trees and genomic similarity matrices were computed using Viptree and Clinker for synteny analysis [51].

## 3. Results

### 3.1. Genome Assembly and Strain Phylogenetic Analysis

The total genome assembly of the strain was 5,814,374 bp with a GC content of 56.84%. The assembly was of high quality, with 0.00 N’s per 100 kbp, indicating no ambiguous bases in the 175 contigs. These metrics collectively demonstrate a robust and continuous genome assembly, suitable for downstream analysis. MLST analysis revealed that the strain belongs to the ST147 high-risk clone according to the Pasteur database.

The phylogenetic investigation (Figure 1) indicates a close evolutionary link between *K. pneumoniae* kpn_R01 and the clinical isolate CP120872 from China. Clinical isolates CP148957 from Japan and CP119564 from the United Kingdom further cluster with this strain, suggesting potential global transmission. The extended family tree includes CP063927, a Chinese environmental isolation, and CP083775, an Australian clinical isolate. The proximity of environmental and clinical isolates indicates that environmental reservoirs contributed to the transmission of the lineage. The identification of analogous strains in China, Japan, the United Kingdom, and Australia contributes to the evidence of potential global dissemination. This may result from infections acquired during hospital care or while patients are abroad.

### 3.2. Antimicrobial Susceptibility and Resistome

*K. pneumoniae* Kpn_R01 is categorized as extensively drug-resistant (XDR) owing to its resistance to multiple antibiotic classes. This classification is supported by minimum inhibitory concentration (MIC) testing, as presented in Table 1, and is further validated through resistance gene analysis utilizing the CARD database. Extensively drug-resistant (XDR) bacteria are defined globally as exhibiting resistance to at least one agent in all but one or two categories of antimicrobials [56]. This strain qualifies for XDR designation as it exhibits resistance to beta-lactams, cephalosporins, carbapenems, monobactams, fluoroquinolones, aminoglycosides, tetracyclines, macrolides, polymyxins, sulfonamides, trimethoprim, phenicols, rifamycins, and fosfomycin. This strain demonstrates resistance to a wide array of antibiotics; however, additional research is required to ascertain whether drugs continue to be effective in addressing pandrug resistance (PDR) criteria. As shown in Table 1, SHV-11 beta-lactamase (*bla*_SHV-11_), NDM-1 carbapenemase (*bla*_NDM-1_), and mutations in porins, encoded by *OmpK37* and *MdtQ*, mediate the strain resistance to beta-lactam antibiotics, resulting in minimum inhibitory concentrations (MICs) of penicillin (≥16 μg/mL), ceftriaxone (≥32 μg/mL), and meropenem (≥8 μg/mL). Aztreonam resistance (≥32 μg/mL) is attributed to low permeability caused by outer membrane porin mutations (*OmpK37*) and efflux (*MdtQ*). Minimum inhibitory concentrations (MICs) of ciprofloxacin (≥4 μg/mL) and levofloxacin (≥8 μg/mL) are primarily due to fluoroquinolones being augmented by efflux pumps (*oqxA*, *acrA*, *marA*) and target mutations (*rsmA*). Enhanced aminoglycoside resistance is ascribed to 16S rRNA methyltransferase (*armA*) and aminoglycoside-modifying enzymes (*aadA2*), leading to minimum inhibitory concentrations (MICs) of ≥16 μg/mL for gentamicin and ≥64 μg/mL for amikacin. Resistance to tetracycline (≥16 μg/mL) and tigecycline (≥2 μg/mL) is facilitated by efflux pumps (*tet(A)* and *KpnF*). Efflux pumps (*KpnF*, *KpnG*, *KpnH*) and enzymatic inactivation (*msrE* and *mphE*) promote resistance to macrolide, resulting in minimum inhibitory concentrations (MICs) of ≥8 μg/mL for erythromycin and ≥16 μg/mL for azithromycin. Resistance to colistin and polymyxin B (≥4 μg/mL) is associated with phosphoethanolamine transferase (*ArnT* and *eptB*) and reduced permeability (*OmpA*). Increased resistance to sulfamethoxazole (≥256 μg/mL) and trimethoprim (≥16 μg/mL) is ascribed to target replacement (*sul1* and *dfrA12*). Fosfomycin resistance (≥64 μg/mL) arises from enzymatic inactivation (*fosA5*). Chloramphenicols, rifamycins, and quaternary ammonium compound resistance is primarily due to the efflux pump. The extensive resistance profile of this strain highlights the challenges in managing infections caused by XDR *K. pneumoniae* and underscores the need for alternative treatment strategies.

### 3.3. Genes Linked to Virulence

Table 2 shows that *K. pneumoniae* Kpn_R01 strain harbors a number of important virulence factors that were identified via the examination of virulence-associated genes. These factors included genes linked to the production of yersiniabactin, enterobactin, and the *E. coli* common pilus (ECP). High coverage (98.31% to 100%) and identity (85.28% to 90.07%) of the ECP gene cluster (*ecpR*, *ecpA*, *ecpB*, *ecpC*, *ecpD,* and *ecpE*) indicates that the strain may manufacture *E. coli* common pili, which are important in adhesion and biofilm formation. This strain has the ability to produce yersiniabactin, a siderophore that aids in iron acquisition and virulence; the yersiniabactin biosynthetic gene cluster (*fyuA*, *ybtE*, *ybtT*, *ybtU*, *irp1*, *irp2*, *ybtA*, *ybtP*, *ybtQ*, *ybtX,* and *ybtS*) was also detected with 100% coverage and 97.62% to 99.95% identity. The strain has the ability to produce enterobactin, a siderophore that plays a crucial role in iron absorption. Genes involved in its production were identified (*fepC*, *fepG*, *entB,* and *entA*), and the coverage and identity were (88.72% to 99.33%) and (80.00% to 82.65%), respectively. In addition to being discovered with 100% coverage and 83.75% identity, the *ompA* gene codes for the outer membrane protein A; hence, this strain may be able to elude the immune system and cause diseases. Among these outcomes are the strain’s pathogenicity-critical capabilities—its ability to adhere, absorb iron, and evade the immune system.

### 3.4. K- and O Serotypes

According to the results of the Kaptive analysis shown in Appendix A, there are two distinct loci that correspond to the K64 and O2a serotypes, respectively: KL64 and O1/O2v1.

With 100% identity and 90.10% coverage, indicating a partial match, the KL64 locus was determined to be typeable. Out of the 24 anticipated genes in the KL64 locus, 23 were identified (95.83%), with 1 gene (KL64_12_wcoT) absent. Numerous genes within the KL64 locus were either partly shortened or incomplete, including *wzx*, *wcoV*, *wzy*, *wcoU*, and *wcsF*, potentially impacting the functioning of the capsular polysaccharide production pathway. Furthermore, three other genes outside the locus were found, namely, *rfaG*, *gmd*, and *HG290*, which are linked to other serotypes (KL40, KL150, and KL4).

The O1/O2v1 locus indicated that the strain was accurately categorized as O2a, exhibiting 99.03% identity and 100% coverage. The seven O-locus genes (*wzm*, *wzt*, *wbbM*, *glf*, *wbbN*, *wbbO*, and *kfoC*) were predicted as present and undamaged with high identity and coverage ranging from 97.31% to 100%. In addition to the locus, six other genes were identified: *manC*, *manB*, *rfbB*, *rfbD*, *rfbA*, and *rfbC*. These genes are often associated with different serotypes, such as O3b, O12, and OL13. These results indicate a highly preserved O2a serotype with possible genetic interchange with other serotypes. The strain exhibits a well-preserved O2a serotype along with supplementary genetic components that may enhance its antigenic variety and indicate a possible genetic exchange with other serotypes.

### 3.5. Plasmids

Several plasmid replicons and colicin genes were found in the strain’s examination of plasmid-associated genes, suggesting that the plasmid content is varied (Table 3). A plasmid linked to NDM carbapenemase resistance may be present in the IncHI1B_1_pNDM-MAR replicon, which was found with 100% coverage and identity. Also present were IncFIB (K)_1_Kpn3 and IncFIB (pKPHS1)_1_pKPHS1 replicons, which are often associated with *K. pneumoniae,* with 100% coverage and 95.54% and 91.07% identity, respectively. The presence of high coverage (100%) and identity (88.60% to 100%) for colicin genes, such as Col440I_1, Col (BS512) 1, and ColpVC_1, indicates that this strain may be capable of producing colicins, bacteriocins that may suppress competing bacterial strains. These results demonstrate that this strain might be capable of horizontal gene transfer and the spread of antibiotic resistance by plasmids.

### 3.6. CRISPR-Cas Systems

The CRISPR-Cas Finder analysis revealed unique CRISPR arrays in two sequences, Seq31 and Seq37. Seq31 consisted of a single CRISPR array (Seq31_1) with a length of 139 bp, featuring 50 bp repeats and 40 bp spacers, demonstrating 98% conservation in both the repeats and spacers. Seq37 exhibited a significantly larger CRISPR array (Seq37_1) of 2649 bp, consisting of 29 bp repeats and 43 spacers, with a repeat conservation of 75.86% and a spacer conservation of 95.69%. Seq37 was oriented in the forward direction and exhibited an evidence level of 4, indicating a well-defined CRISPR locus. The findings demonstrate that although CRISPR elements are present in the dataset, their prevalence is low, with only a limited number of sequences showing functional or well-conserved CRISPR arrays (Appendix A).

### 3.7. Prophage Regions and Viral Sequences

VirSorter analysis detected 14 potential prophage sequences in the sample, the majority of which were dsDNA phages, as well as one ssDNA phage (Seq129). The prophage sequences varied in length from 2300 bp (Seq129) to 49,050 bp (Seq34). The highest confidence values (maximum score = 1.00) were found in Seq34 and Seq43, indicating strong prophage signals. The number of signature genes ranged from 0 to 14, with Seq34 having the most. Multiple sequences, including Seq86 and Seq102, had low confidence scores (0.613 and 0.50, respectively) and lacked hallmark genes, indicating weaker prophage signatures. The discovery of these putative prophages provides information about potential viral components inside the investigated bacterial genomes, which may influence bacterial evolution and resistance mechanisms (Appendix A).

### 3.8. Klebsiella Phage Kpn_R1 Isolation, Morphology, Assembly and Taxonomy

The phage was isolated from hospital wastewater with distinct plaque morphology (Figure 2A). TEM image examination showed that *Klebsiella* phage Kpn_R1 had a polyhedral head with a diameter of 53.7 nm and a wavy 178.3 nm long tail morphology (Figure 2B). This, pursuant to the International Committee on Taxonomy of Viruses taxonomy system, places them in the *Demerecviridae* family.

### 3.9. General Genome Characteristics

The phage genome is composed of 172,025 bp base pairs (bp), with a GC content of 46.3%. The genome consists of 211 predicted genes, with an average gene length of 639.47 bp. The genome type is classified as unknown, and it is of high quality with a 100.0% genome quality score with 100.0% completeness and 0.0% contamination.

The probability that the phage is temperate was estimated at 81%, based on the BACPHLIP database. No antibiotic resistance genes were identified within the genome, indicating that the phage does not haror any known antibiotic resistance traits, and the phage is confirmed not to be a provirus. The linearity of the phage genome was confirmed by PhageTerm. The genome consists of 220 predicted genes, with an average gene length of 639.47 bp. The phage genome organization is shown in Figure 3.

### 3.10. Predicted Host Range, Evolutionary Relationships, and Genomic Similarity of Klebsiella Phage Kpn_R1

The predicted hosts of the annotated phage were inferred based on the best-matching phage genomes and their reported hosts. As shown in Table 4, the majority of hits (33 hits) were associated with *Klebsiella*, particularly *K. pneumoniae,* with an average genetic distance of 0.06, an average matching hash score of 215.3, and an average *p*-value of hits 3.38 × 10^−111^. This strong association suggests that *K. pneumoniae* is the most likely host for the annotated phage. Other potential hosts include *Yersinia* (2 hits), *Salmonella* (13 hits), *Erwinia* (1 hit), and *Campylobacter* (1 hit), though these associations are weaker, as indicated by higher genetic distances (0.22–0.23).

Table 5 shows that the analysis grouped the closest relatives into several genera, with the highest number of hits belonging to *Sugarlandvirus* (20 hits) and *Epseptimavirus* (15 hits). The *Sugarlandvirus* genus showed the lowest average genetic distance (0.04) and the highest average matching hashes (250.8), indicating a strong genomic similarity. These results suggest that the isolated phage belongs to *Caudoviricetes*; *Demerecviridae*; *Sugarlandvirus* genus.

The evolutionary relationships of *Klebsiella* phage Kpn_R1 were investigated using whole-genome sequencing comparisons (Figure 4). The circular phylogenetic tree shows different clusters of closely related phages, with Kpn_R1 located next to *Klebsiella* phage vB_Kpn_IME260 and *Klebsiella* phage Sugarland. Kpn_R1 shares 92.8% genomic identity across 31.5% of its sequence with vB_Kpn_IME260, as well as 92.3% identity over 30.1% of its genome with *Klebsiella* phage Sugarland. The study revealed Escherichia phage N15, which had a mean identity of 69.8% across 14% of the genome, indicating a significant evolutionary difference.

The alignment of protein sequences (Figure 5) further highlights the similarities between Kpn_R1 and its closest relatives. The strongest sequence similarities between *Klebsiella* phage Kpn_R1 (Seq1), *Klebsiella* phage vB_Kpn_IME260, and *Klebsiella* phage Sugarland are depicted by connected syntenic blocks, with the most conserved regions shown in pink. The distinct structural organization of *Escherichia* phage N15 further reinforces its evolutionary divergence.

As shown in the Appendix A, the whole-genome sequencing of the isolated phage demonstrated a 92.8% identity across 31.5% of the whole genome of its closest similar *Klebsiella* phage vB_Kpn_IME260. *Klebsiella* phage Sugarland (NC_042093) exhibited a mean identity of 92.3% across 30.1% of the genome length, indicating a modest degree of similarity with related phages. Sugarland is classified within the Pseudomonadota category, signifying a wider host range among Enterobacteriaceae members.

The comparison also recognized *Escherichia* phage N15 (NC_001901) as a similar phage; however, it had a lower mean identity of 69.8% across merely 14% of the genome. Different from lytic phages such as Sugarland, N15 is a completely defined temperate phage that may integrate into the bacterial genome as a linear plasmid. The somewhat low identity and genome coverage suggest that N15 is evolutionarily different even if it shares certain genetic elements with the isolated phage. These findings suggest that the isolated phage may adapt its genome for better survival or bacterial genome evolution via horizontal gene transfer technologies due to its evolutionary links and functional capacity. Comparative genomics will dominate future research to identify distinct genetic variables influencing host specificity and lytic activity.

### 3.11. Klebsiella Phage Kpn_R1 Genome Annotation

The 220 ORFs (Appendix A) in the annotated phage genome have been categorized into functional categories based on their anticipated functions. As shown in Table 6, hypothetical/uncharacterized proteins comprise the majority of the total ORFs, accounting for 48.34% (102 ORFs). These proteins are not known to have any functions or exhibit any significant similarities to known proteins. This implies that there are novel genes that are unique to phages and may play distinct functions in the phage lifecycle. Structural proteins constitute the second-largest group, which accounts for 18.01% (38) of the ORFs. This encompasses the primary capsid protein, tail fiber proteins, and portal proteins, all of which are crucial for the formation of virion structures. The significant presence of structural proteins highlights an accurate virion assembly for effective infection and replication. Proteins associated with DNA replication and metabolism comprise 11.85% (25) of the open reading frames (ORFs). DNA polymerase, helicase, exonuclease, and recombination mediators are all in this group. These components are essential for the preservation and replication of genomes. The phage’s ability to modify host DNA and facilitate effective genome replication is emphasized by the presence of these proteins. Transcriptional regulators, anti-termination proteins, and XRE family regulators comprise 2.84% (6) of the ORFs. It is probable that these proteins are involved in the regulation of gene expression throughout the phage lifecycle, which allows for the rapid activation or suppression of genes that are necessary for the various stages of infection.

The host lysis group, which comprises endolysin, holin, and spanin proteins, comprises 1.42% (3) of the ORFs. These proteins are crucial for lysing the host cell to liberate freshly formed phage particles, signifying the concluding phase of the lytic cycle. The category of Moron, auxiliary metabolic genes, and host takeover comprise 6.16% (13) of the ORFs. Also included in the genome annotated ORFs are the ParA/B-like proteins comprising 1.42% (3). These proteins participate in plasmid partitioning with protelomerase presence, indicating a mechanism for the steady inheritance of the phage genome throughout cellular division, akin to the N15 phage and protelomerase reinforces this similarity since protelomerase is characteristic of N15’s replication and maintenance approach, facilitating telomere resolution and linear genome preservation. The genome encodes a toxin–antitoxin (TA) system, especially a RelE/ParE family toxin (Appendix A). Plasmid maintenance, stress response, and persistence mechanisms are frequently implemented by TA systems. In phages, they may influence the behavior of host cells to promote phage survival or facilitate the stability of the prophage state. The phage genome exhibits a functional distribution that is typical of temperate phages, with a focus on host lysis, DNA replication, and structural assembly. The extensive unexplored biology is suggested by the significant predominance of putative proteins. However, the presence of partitioning proteins, protelomerase, and a toxin–antitoxin system suggests a reproductive and maintenance strategy similar to that of the N15 phage. In order to confirm the mechanisms of the phage’s lifecycle and to elucidate the functions of the potential proteins, additional experimental investigations are required.

## 4. Discussion

In the current study, *K. pneumoniae* Kpn-R1 strain was isolated from an ICU patient’s sputum sample. The strain’s genome size was 5,835,643 bp, with a GC content of 56.82%, according to whole genome sequencing. The strain was identified as a high-risk ST 147 clone using Pasteur Institute MLST. The genome size of *K. pneumoniae* ST147 isolates is generally between 5.0 and 5.5 Mbp and a GC content range of 56–58% [2]. These genomic traits reflect the species’ flexibility and ability to accept foreign genetic material, such as plasmids and integrons, which typically carry antibiotic resistance genes [90].

Kpn_R01 strain exhibits an XDR profile influenced by 32 AMR spanning multiple antibiotic classes, including beta-lactams, fluoroquinolones, aminoglycosides, tetracyclines, macrolides, phenolics, polymyxins, and sulfonamides. According to Nordmann et al. (2011) and Logan and Weinstein (2017), there is a wide resistance pattern associated with the increasing incidence of hospital-acquired *K. pneumoniae* bacteria that are resistant to several drugs [4,91]. The research revealed that the Kpn_R01 genome contains beta-lactamase resistance genes including *bla*_SHV-11_, *bla*_NDM-1_, *OmpK37*, and *MdtQ*. SHV-11 beta-lactamase and NDM-1 carbapenemase are the primary enzymes for penicillin, cephalosporin, and carbapenem degradation [92]. Alterations in porins (OmpK37) and efflux (MdtQ) impede the entry of beta-lactamase into the bacterial cell by reducing cell permeability [93]. ST147 is a high-risk clone that has been extensively documented and is associated with carbapenem resistance, notably as a result of the production of carbapenemases like NDM-1, SHV-11, and OXA-48 [94]. Studies undertaken in Saudi Arabia, UAE, and Oman have highlighted the frequency of ST147 in clinical settings, with a significant percentage of isolates carrying the *bla*_NDM-1_ and *bla*_OXA-48_ genes [95,96,97].

Resistance to fluoroquinolones, including ciprofloxacin (≥4 μg/mL) and levofloxacin (≥8 μg/mL), is facilitated by a synergy of efflux pumps (*oqxA*, *acrA*, *marA*). These methods diminish intracellular drug concentrations and modify drug-binding sites, hence imparting significant resistance to this family of antibiotics [98]. The strain’s resistance to aminoglycosides, such as gentamicin (≥16 μg/mL) and amikacin (≥64 μg/mL), is mediated by 16S rRNA methyltransferase (*armA*) and aminoglycoside-modifying enzymes (*aadA2*) [99]. Efflux pumps (*tet(A)*, *tet(B)*), and ribosomal protection proteins (tet(M)) mediate resistance to tetracyclines (tetracycline ≥16 μg/mL, tigecycline ≥2 μg/mL), which reduce intracellular drug concentrations or protect the ribosome from inhibition [89,100]. Similarly, efflux transporters (*msrE*, *mphE*) are mechanisms that have been increasingly observed in Gram-negative bacteria [101] that facilitate resistance to macrolides (erythromycin ≥8 μg/mL, azithromycin ≥16 μg/mL). The strain’s resistance to trimethoprim (≥16 μg/mL) and sulfamethoxazole (≥256 μg/mL) is mediated by dihydropteroate synthase (*sul1*) and dihydrofolate reductase (*dfrA12*), respectively. The drug targets are altered by these enzymes, which prevent the inhibition of folate biosynthesis, a critical pathway for bacterial proliferation, thereby conferring resistance [102]. Similarly, fosfomycin thiol transferase (*fosA5*) is responsible for the inactivation of the antibiotic, resulting in resistance to fosfomycin (≥64 μg/mL) [103]. Phosphoethanolamine transferase (*arnT*, *eptB*) and mutations in the mgrB gene are associated with resistance to colistin and polymyxin B (≥4 μg/mL). These mutations modify the lipid A component of lipopolysaccharide, thereby reducing the binding affinity of polymyxins [104]. This mechanism is particularly alarming as polymyxins are frequently employed as last-resort antibiotics to treat XDR Gram-negative infections. The efflux transporters (*qacEdelta1*, *leuO*) that are frequently associated with biocide resistance in Gram-negative bacteria facilitate the strain’s resistance to quaternary ammonium compounds (≥50 μg/mL) [105]. The strain’s persistence in hospital environments, where biocides are frequently employed for disinfection, may be influenced by this resistance mechanism [74].

In addition to its extensive resistance profile, the strain possessed several virulence factors that exacerbated its pathogenesis. The *E. coli* common pilus (ECP) gene cluster implicates the strain’s capacity to adhere to host tissues and form biofilms, which is a critical factor in the establishment of infections [106]. Enterobactin genes and the yersiniabactin biosynthetic gene cluster suggest that the strain is capable of acquiring iron, a critical nutrient for bacterial survival and virulence in the host [107]. Outer membrane protein A has been demonstrated to safeguard microorganisms from host immune responses [108], which further supports the notion that the *ompA* gene has immune evasion capabilities. These virulence factors collectively contribute to the strain’s capacity to cause severe infections, particularly in immunocompromised patients.

The strain has a non-operational Type I-E CRISPR-Cas gene cluster, diminishing its tolerance to foreign genetic elements, such as plasmids and phages, hence facilitating horizontal gene transfer (HGT) and the dissemination of antibiotic resistance genes [109]. This property is characterized by its plasmid content, including In-cHI1B_1_pNDM-MAR, IncFIB (K)_1_Kpn3, and IncFIB (pKPHS1)_1_pKPHS1 replicons [110]. The strain’s ability to generate bacteriocins is shown by the presence of colicin genes (Col440I_1, Col (BS512)_1, ColpVC_1), potentially augmenting its survival in polymicrobial settings and suppressing the proliferation of rival bacterial strains [111]. Eleven prophage regions were also detected, which often harbor genes that enhance bacterial virulence and antibiotic resistance, highlighting the genetic diversity of strains and their potential for horizontal gene transfer [5]. Eleven prophage regions were also detected, which often harbor genes that enhance bacterial virulence and antibiotic resistance, highlighting the genetic diversity of the strain and its potential for horizontal gene transfer [8]. This property makes this Middle Eastern ST147 a uniquely adaptable nosocomial pathogen.

Of particular interest is the isolation of an N15-like temperate phage from hospital wastewater using the Kpn-R1 strain as a host. This study’s significant development is the characterization of a N15-like temperate phage exhibiting a linear plasmid prophage lifestyle [9], a replication method seldom reported in *K. pneumoniae* phages. This phage, in contrast to other phages often linked to ST147, utilizes protelomerase (93.3% similarity to N15 TelN) and ParA/B partitioning proteins to sustain lysogeny, resembling the N15 phage’s capacity to exist as a linear plasmid [9,11,112]. The incorporation of a RelE/ParE toxin–antitoxin (TA) system within the phage genome further differentiates it from previous accounts of N15-like pKO2 linear plasmid prophage of *K. oxytoca* [14]. The presence of TA systems, usually associated with plasmid stability, indicates a unique function in phage-mediated stress adaptation and prophage survival [113,114]. These results enhance the understanding of phage roles in bacterial evolution, especially in high-risk clones such as ST147 where prophages may serve as repositories for resistance or virulence genes.

The structural proteins identified in the genome, including major capsid proteins, tail fibers, and portal proteins, are essential for virion assembly and host recognition. These proteins are highly conserved among phages and play a critical role in the infection cycle [115,116]. The regulatory proteins identified in the genome, including transcriptional regulators and anti-termination proteins, ensure the timely activation or repression of genes required for different stages of infection, such as lysogeny and lytic replication [117]. The UmuD-like protein identified in the genome is part of the SOS response system, which is involved in DNA damage repair and mutagenesis [118,119]. This finding is consistent with studies on other phages where UmuD homologs have been shown to enhance survival in hostile environments [88].

This study uniquely associates CRISPR inactivation with MGE-driven evolution in Middle Eastern clones, a feature previously linked to plasmid loss in other lineages [89]. Furthermore, we provide the first evidence of an N15-like phage containing TA systems in ST147, contesting the notion that phage-mediated adaptation in *K. pneumoniae* is restricted to lytic or temperate phages. These results reclassify ST147 as a paradigm for investigating region-specific pathogen evolution and phage–host coadaptation.

This study bridges a critical gap by characterizing the first N15-like phage within the high-risk ST147 clone, revealing its unique TA system and replication strategy. By linking clinical and environmental reservoirs, we propose actionable strategies for phage engineering and surveillance to combat XDR *K. pneumoniae* in healthcare settings. These findings shift the paradigm from observational genomics to translational solutions targeting phage–bacterial coevolution.

## 5. Conclusions

This study presents the genomic interaction between a novel N15-like temperate phage sourced from hospital wastewater and an extremely drug-resistant *K. pneumoniae* ST147 strain isolated from an ICU patient, thereby revealing significant new insights into resistance dissemination and phage-mediated evolution. The ST147 strain’s hybrid plasmid (IncHI1B_1_pNDM-MAR/IncFIB), which carries *bla*_NDM-1_ and colicin genes, along with CRISPR-Cas inactivation and 11 prophage regions, highlights its genetic plasticity and potential nosocomial threat. The linear plasmid-like replication system of the phage, along with protelomerase and a novel RelE/ParE toxin–antitoxin (TA) system—previously unreported in temperate *Klebsiella* phages—highlights mechanisms that enhance resistance traits and promote phage survival.

Tracking phage-mediated resistance gene transfer between clinical and environmental reservoirs helps us show how wastewater monitoring may prevent outbreaks. By suggesting actionable plans, this work goes beyond the current genomic cataloging of ST147: the phage’s replication machinery provides a template for engineered therapies (e.g., CRISPR-Cas delivery), while the strain’s regional resistance profile calls for tailored antibiotic stewardship to reduce last-resort drug misuse. Emphasizing phage–bacterial interactions as a frontier in the fight against XDR infections, our results change the paradigm from passive genomic monitoring to proactive, ecology-driven therapies. Future research should confirm the functional significance of the TA system and investigate phage-based diagnostics to interfere with resistance transfer in healthcare environments.

## Figures and Tables

**Figure 1 microorganisms-13-00908-f001:**
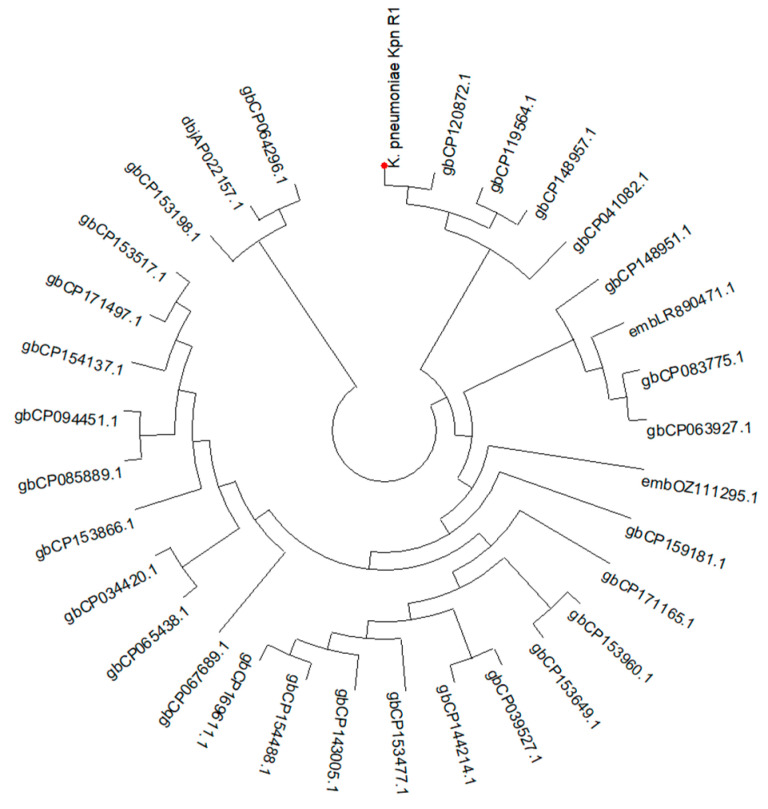
A circular phylogenetic tree illustrating the evolutionary relationships of *K. pneumoniae* Kpn_R01 and its 34 closest strains. The tree was inferred from 16S rRNA gene sequences using the maximum likelihood (ML) tree highlighting the phylogenetic placement of Kpn_R01 (marked with a red dot) among closely related strains. Clustering patterns indicate potential global dissemination, with strains originating from diverse geographic locations. The close association between clinical and environmental isolates underscores the role of environmental reservoirs in bacterial transmission. The tree was inferred using the ML method with the Tamura–Nei model [52], and branch support was assessed with 500 bootstrap replicates [53]. The initial heuristic search compared a neighbor-joining (NJ) tree [54] and a maximum parsimony (MP) tree, selecting the one with the best log-likelihood score. Evolutionary analyses were conducted in MEGA12 [55].

**Figure 2 microorganisms-13-00908-f002:**
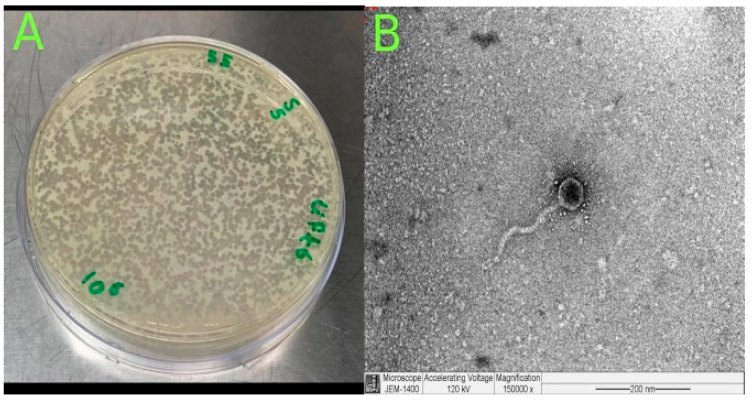
Isolation and characterization of *Klebsiella* phage Kpn_R1. (**A**) Plaque morphology of *Klebsiella* phage Kpn_R1 on a bacterial lawn, showing distinct, clear plaques. (**B**) Transmission electron microscopy (TEM) image of *Klebsiella* phage Kpn_R1, revealing a polyhedral head (53.7 nm in diameter) and a wavy tail (178.3 nm in length). Based on morphology, the phage is classified within the *Demerecviridae* family according to the International Committee on Taxonomy of Viruses (ICTV). Scale bar: 200 nm.

**Figure 3 microorganisms-13-00908-f003:**
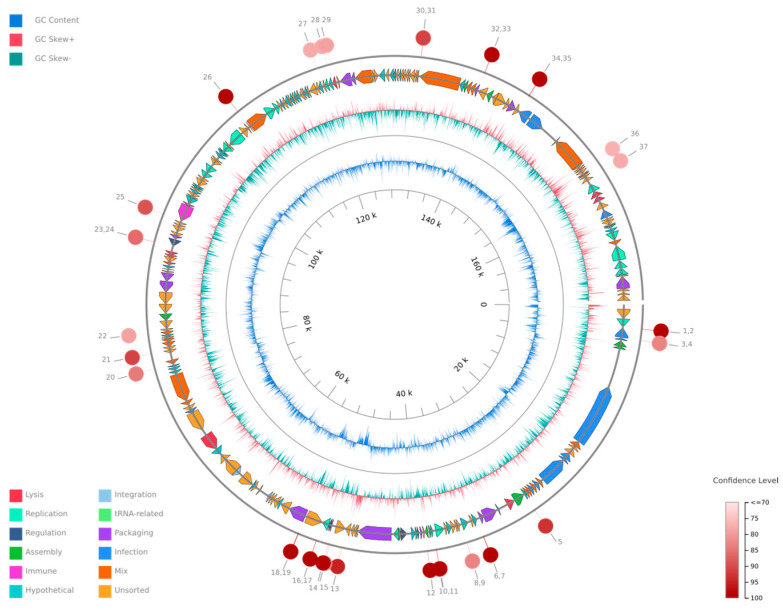
A circular genome map of *Klebsiella* phage Kpn_R1 generated using PhageScope. The outermost rings represent annotated coding sequences (CDSs) classified by function, while the inner rings display GC content and GC skew. Red markers indicate putative virulence-associated or antibiotic resistance-related genes. The color gradient represents the variation in GC content across the genome.

**Figure 4 microorganisms-13-00908-f004:**
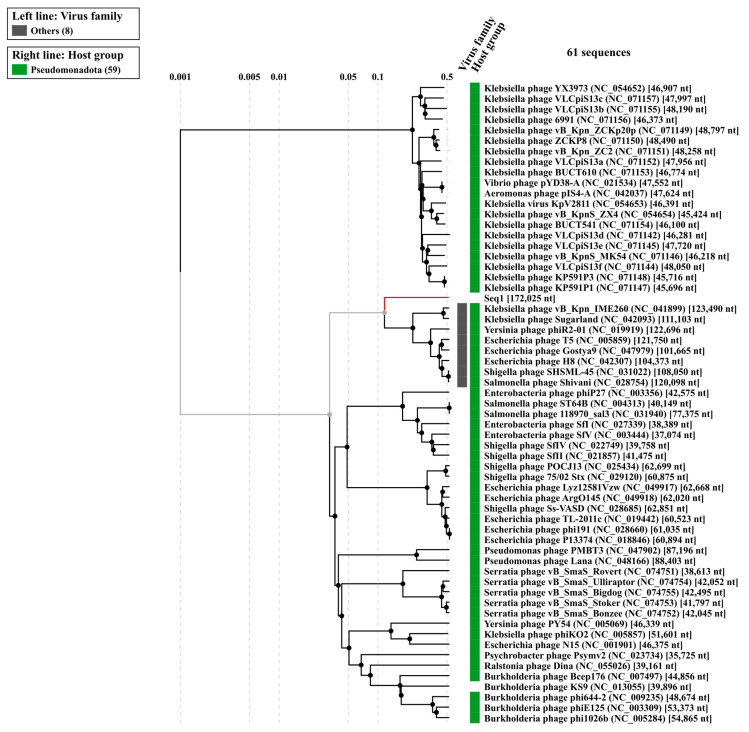
The phylogenetic analysis of *Klebsiella* phage Kpn_R1 was conducted using ViPTree, a web server that generates viral proteomic trees based on genome-wide sequence similarities computed by tBLASTx. The circular proteomic tree illustrates the evolutionary relationships of Kpn_R1, highlighting its closest related phages forming distinct clusters (gray).

**Figure 5 microorganisms-13-00908-f005:**
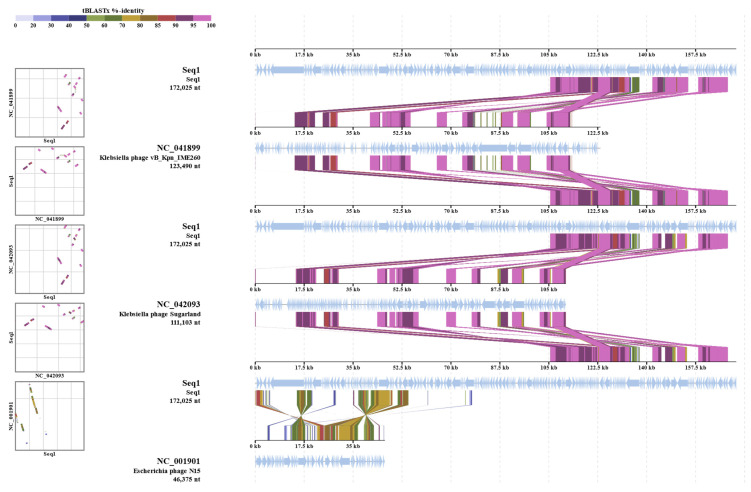
Proteomic similarity map of *Klebsiella* phage Kpn_R1 and related phages. The alignment shows protein sequence similarities between *Klebsiella* phage Kpn_R1 (Seq1) and its closest relatives: *Klebsiella* phage vB_Kpn_IME260, *Klebsiella* phage Sugarland, and *Escherichia* phage N15. Conserved regions are depicted by connected syntenic blocks, with the strongest similarities shown in pink. The distinct structural organization of *Escherichia* phage N15 highlights its evolutionary divergence.

**Table 1 microorganisms-13-00908-t001:** The minimum inhibitory concentrations (MICs) for various antimicrobials and the resistance mechanisms identified in the strain.

Antimicrobial Class	Antimicrobial Agent	MIC (μg/mL)	Gene	Resistance Mechanism	Notes	References
Beta-lactams	Penicillin	≥16	*bla*_SHV-11_ *bla*_NDM-1_ *OmpK37*, *MdtQ*	Beta-lactamase carbapenemase porin mutation	High resistance due to beta-lactamases and reduced permeability.	[57,58,59,60]
Cephalosporin	Ceftriaxone	≥32				
Carbapenem	Meropenem	≥8	*bla*_NDM-1_ *OmpK37*, *MdtQ*	Carbapenemase porin mutation	Carbapenem resistance due to carbapenemase and reduced permeability	[27,28,29]
Monobactam	Aztreonam	≥32	*OmpK37*, *MdtQ*	Porin mutation	Resistance due to reduced permeability	[28,29]
Fluoroquinolones	Ciprofloxacin	≥4	*oqxA*, *acrA*, *marA*, *rsmA*	Efflux pumps	High resistance due to efflux and target mutations	[61,62,63,64]
	Levofloxacin	≥8				
Aminoglycosides	Gentamicin	≥16	*armA* *aadA2*	16S rRNA methyltransferase aminoglycoside-modifying enzyme	High-level resistance due to target modification and enzymatic inactivation	[65]
	Amikacin	≥64	*armA*	16S rRNA methyltransferase	Resistance due to target modification	[65]
Tetracyclines	Tetracycline	≥16	*tet(A)*, *K. pneumoniae KpnF*	Efflux pump	Resistance due to efflux-mediated resistance	[66,67]
	Tigecycline	≥2				
Macrolides	Erythromycin	≥8	*msrE* *mphE* *KpnF*, *KpnG*, *KpnH*	ABC-F protein macrolide phosphotransferase efflux pumps	Resistance due to efflux and enzymatic inactivation.	[60,68,69]
	Azithromycin	≥16				
Peptide Antibiotics	Colistin (Polymyxin E)	≥4	*ArnT*, *eptB* *OmpA*	Phosphoethanolamine transferase Porin	Resistance due to target modification and reduced permeability	[70]
	Polymyxin B	≥4				
Sulfonamides	Sulfamethoxazole	≥256	*sul1*	Sulfonamide-resistant dihydropteroate synthase	High resistance due to target replacement	[71]
Diaminopyrimidines	Trimethoprim	≥16	*dfrA12*	Trimethoprim-resistant dihydrofolate reductase	Resistance due to target replacement	[66]
Phenicols	Chloramphenicol	≥32	*acrA*, *rsmA*	Efflux pumps	Resistance due to efflux mechanisms	[62,64]
Bleomycin	Bleomycin		*ble* _MBL_	BRP_MBL_ (Bleomycin Resistance Protein)	Resistance due to target alteration	[72]
Rifamycins	Rifampin	≥8	*KpnF*, *KpnE*, *marA*	Efflux pumps	Resistance due to efflux mechanisms	[60]
Fosfomycin	Fosfomycin	≥64	*fosA5*	Fosfomycin thiol transferase	Resistance due to enzymatic inactivation	[73]
Disinfectants/ Antiseptics	Quaternary ammonium compounds	≥50	*qacEΔ1*, *leuO*	Efflux pump, transcription regulator	Resistance due to efflux mechanisms	[74]

**Table 2 microorganisms-13-00908-t002:** Virulence-associated genes identified in the strain.

Gene	% Coverage	% Identity	Product	Virulence Factor	Accession
*ecpR*	98.31	86.58	regulator protein [EcpR]	*E. coli* common pilus (ECP)	NP_286011
*ecpA*	99.32	90.07	*E. coli* common pilus structural subunit [EcpA]		NP_286010
*ecpB*	100.00	87.89	*E. coli* common pilus chaperone [EcpB]		NP_286009
*ecpC*	99.96	87.42	*E. coli* common pilus usher [EcpC]		NP_286008
*ecpD*	100.00	88.81	polymerized tip adhesin of ECP fibers [EcpD]		NP_286007
*ecpE*	99.74	85.28	*E. coli* common pilus chaperone [EcpE]		NP_286006
*fyuA*	100.00	99.95	pesticin/yersiniabactin receptor protein [FyuA]	Yersiniabactin biosynthesis	NP_405467
*ybtE*	100.00	99.75	siderophore biosynthetic protein [YbtE]		NP_405468
*ybtT*	100.00	99.75	Yersiniabactin [YbtT]		NP_405469
	100.00	99.73	Yersiniabactin [YbtU]		NP_405470
*irp1*	100.00	99.82	Yersiniabactin [Irp1]		NP_405471
*irp2*	100.00	99.75	Yersiniabactin [Irp2]		NP_405472
*ybtA*	100.00	99.79	Yersiniabactin transcriptional regulator [YbtA]		NP_405473
*ybtP*	100.00	99.72	lipoprotein inner membrane ABC-transporter [YbtP]		NP_405474
*ybtQ*	100.00	99.89	inner membrane ABC-transporter [YbtQ]		NP_405475
*ybtX*	100.00	97.74	putative signal transducer [YbtX]		NP_405476
*ybtS*	100.00	97.62	salicylate synthase [Irp9]		NP_405477
*fepC*	94.61	81.22	ferrienterobactin ABC transporter ATPase [FepC]	Enterobactin biosynthesis	NP_752606
*fepG*	88.72	80.07	iron-enterobactin ABC transporter permease [FepG]		NP_752607
*entB*	99.18	82.65	Isochorismatase [EntB]		NP_752613
*entA*	99.33	80.00	23-dihydro-23-dihydroxybenzoate dehydrogenase [EntA]		NP_752614
*ompA*	100.00	83.75	outer membrane protein A [OmpA]	Immune evasion	AAF37887

**Table 3 microorganisms-13-00908-t003:** Plasmids predicted from whole genome sequencing (WGS) contigs of *K. pneumoniae* Kpn_R01 strain using PlasmidFinder (v2.1.6).

Strand	Plasmid	Coverage %	Identity %	Product	Accession/References
+	Col440I_1	100	96.49	Col440I_1__CP023920.1	CP023920.1 [75]
−	IncHI1B_1_pNDM-MAR	100	100.00	IncHI1B_1_pNDM-MAR_JN420336 (NDM carbapenemase)	JN420336 [76]
+	Col(BS512)_1	100	100.00	Col(BS512)_1__NC_010656	NC_010656 [77]
−	ColpVC_1	100	88.60	ColpVC_1__JX133088	JX133088 [78]
−	IncFIB(pKPHS1)_1_pKPHS1	100	95.54	IncFIB(pKPHS1)_1_pKPHS1_CP003223	CP003223 [79]
+	IncFIB(K)_1_Kpn3	100	91.07	IncFIB(K)_1_Kpn3_JN233704	JN233704 [80]

**Table 4 microorganisms-13-00908-t004:** Predicted hosts as inferred through the best-matching phage genomes and their reported hosts.

Species	Hits	Average Matching Hashes	Average *p*-Value of Hits	Average Genetic Distances
* Erwinia *	1 hit	5	4.36 × 10^−27^	0.22
*Yersinia*	2 hits	5	5.09 × 10^−27^	0.22
*Klebsiella*	33 hits	215.3	3.38 × 10^−111^	0.06
*Salmonella*	13 hits	4.15	1.21 × 10^−21^	0.23
*Campylobacter*	1 hit	4	1.50 × 10^−21^	0.23

**Table 5 microorganisms-13-00908-t005:** Taxonomic classification and genetic similarity metrics of *Klebsiella* phage Kpn_R1 and its closest relative genera (using the MASH algorithm), All hits with genetic distance < 1 against published phage genomes grouped by genus.

Genus	Hits	Average Matching Hashes	Average *p*-Value of Hits	Average Genetic Distances
Unclassified	14 hits	149.57	3.11 × 10^−28^	0.1
*Epseptimavirus*	15 hits	4.27	1.13 × 10^−21^	0.23
*Sugarlandvirus*	20 hits	250.8	0.00 × 10^+00^	0.04
*Loughboroughvirus*	1 hit	4	2.31 × 10^−22^	0.23

**Table 6 microorganisms-13-00908-t006:** Functional categorization of *Klebsiella* Phage Kpn_R1 open reading frames (ORFs).

Category	Number of Proteins	Percentage	Key Examples	References
Partioning and maintainance	3	1.42	SopA/ParB proteins	[81]
Unknown	102	48.34	Proteins labeled as “hypothetical”, “uncharacterized”, or with DUF domains	-
Tail, head, connector, and packaging	38	18.01	Tail, tail fibers, tail tape measure, Terminase (TerL/S), scaffolding protein, prohead protease, capsid, head, host division inhibitor, connector	[82]
DNA, RNA, and nucleotide metabolism	25	11.85	DNA polymerase, helicase, ribonucleotide reductase, exonuclease, thymidylate synthase	[83]
Transcription regulation	6	2.84	Cro/CI regulator, LexA, antitermination protein Q, sigma-54 factor, DNA methyltransferase, receptor-binding protein	[84]
Lysis	3	1.42	Holin, endolysin	[85]
Moron, auxiliary metabolic genes, and host takeover	13	6.16	Glutamate 5-kinase, PhoH-like protein, thioredoxin, tRNA-like domains	[86,87]
Integration and excision	2	0.95	Integrase, excisionase Xis	[88]
others	18	8.53	-	-
Protelomeras	1	0.47	Protelomeras	[89]
Total	211	100	-	-

## Data Availability

The original contributions presented in this study are included in the article/Appendix A. Further inquiries can be directed to the corresponding author.

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
