# Peer review of "Comprehensive Genomic Analysis of Klebsiella pneumoniae and Its Temperate N-15-like Phage: From Isolation to Functional Annotation"

_microorganisms, 2025, doi:10.3390/microorganisms13040908_

Round 1
Reviewer 1 Report
Comments and Suggestions for Authors
comments
The manuscript present interesting information but I considered that many results are excessively descriptive and sometimes repetitive. Authors need to focus the results to present them in a more directed view to highlight the relevant insights. In addition, some tables can be included as supplementary data and only include those that are considerably important to explain. Maybe include a Table summarizing phage and bacteria genome features could be desirable to avoid excessive description in the text.
Introduction has information but it is unstructured and did not provide a theoretical framework listing information that seems discussion or describe the results.
Delete short title
Line 22, change E coli in italics
lines 35-44, paragraph too long, please resume for the abstract
Author summary is not required.
Please use the same font along the text. In some lines the format is distinct.
line 64, please avoid contractions and along the text, this is informal
line 76, the verb mix is not clear to mention phage integration
line 132-133 and 285-286, information of bioproject and accession numbers should be provided in material and methods.
Figure 1, phylogenetic tree was done with 16S rRNA sequence or complete genome?
Results regarding MIC are too long, please show the most important results.
Table 3, edit genes in italics
lines 289-292, information is repetitive.
information of phylogenrtics analyses is repetitive.
line 381, what were the criteria to define morons, AMG and host takeover?
In methods authors mention phageTerm analysis but the results are not described in the text
In discussion, many results are mentioned but it is poorly discussed about the biological significance, and I considered a reestructuration of discussion is needed.
line 603, what is the source of Klebsiella strains? can authors provide citation?
line 612-624, what type of library was done? 2 x 150 bp?, what were the trimmomatic parameters for trimming?
5.4 and 5.5 section can be merge and avoid repetitive information.
sections 5.4, 5.5 and 5.6 have mix information, I suggest to mention a section specific for DNA isolation and sequencing for bacteria, and other section for phage.
Section 5.7 is long but no specific information about statistical tests is provided.
Improved the format of tables. Extensive tables can be included as supplementary information, not in the manuscript.
Author Response
Comment |
Introduction has information but it is unstructured and did not provide a theoretical framework listing information that seems discussion or describe the results. |
Response |
Thank you for your important comment, the introduction part has been restructured in the main manuscript. |
Comment |
Delete short title |
Response |
Thank you for your clarification. Short titles are not permissible, so we have removed them accordingly. |
Comment |
Line 22, change E coli in italics |
Response |
Thank you for your notice, we have italicized all scientific names through the main manuscript. |
Comment |
lines 35-44, paragraph too long, please resume for the abstract. |
Response |
We sincerely appreciate your insightful feedback. As suggested, we have revised the abstract to ensure conciseness while emphasizing the study’s objectives and key findings. |
Comment |
Author summary is not required. |
Response |
Thank you for your clarification. Author summary are not permissible, so we have removed them accordingly. |
Comment |
Please use the same font along the text. In some lines the format is distinct.
|
Response |
Thank you for your observation, the fonts were adjusted in the manuscript. |
Comment |
line 64, please avoid contractions and along the text, this is informal |
Response |
Thank you for this correction, the sentence was replaced by formal form: The capacity of K. pneumoniae to acquire and disseminate antibiotic resistance genes through horizontal gene transfer routes, such as plasmids, transposons, and prophages, has significantly contributed to its reputation as a significant nosocomial pathogen. |
Comment |
line 76, the verb mix is not clear to mention phage integration: |
Response |
Thank you for your notice, the word mix was changed to integrate. |
Comment |
line 132-133 and 285-286, information of bioproject and accession numbers should be provided in material and methods. |
Response |
Thank you for your comment, the information’s of bioproject and accession numbers were removed to material and methods. |
Comment |
Figure 1, phylogenetic tree was done with 16S rRNA sequence or complete genome? |
Response |
Thank you for raising this important point. The phylogenetic tree in Figure 1 was constructed using 16S rRNA gene sequences, not complete genomes. The caption has now been revised to explicitly state this, ensuring clarity for readers. While 16S rRNA is a widely used marker for phylogenetic analysis at the species level, we acknowledge its limitations in resolving fine-scale differences among closely related strains within Klebsiella pneumoniae. To strengthen our interpretation, we emphasize that the observed clustering patterns align with geographic and clinical/environmental metadata, supporting the plausibility of the inferred relationships. Future studies incorporating whole-genome sequencing or multi-locus sequence typing (MLST) could further refine these phylogenetic insights. The methodological details, including the use of the Tamura-Nei model and bootstrap validation, remain appropriate for 16S rRNA-based analyses. |
Comment |
Results regarding MIC are too long, please show the most important results. |
Response |
Thank you for your comment, antibiotic data were reduced to the most important ones. |
Comment |
Table 3, edit genes in italics. |
Response |
Thank you for your observation, all gene names through the manuscript were revised and italicized. |
Comment |
lines 289-292, information is repetitive. |
Response |
Thank you for your observations. Those lines were removed. |
Comment |
information of phylogenetic analyses is repetitive. |
Response |
Thank you for your feedback. We have streamlined the text to eliminate redundancy in the description of Figure 4. The caption is now as follows: Figure 4. The phylogenetic analysis of Klebsiella phage Kpn_R1 was conducted using ViPTree, a web server that generates viral proteomic trees based on genome-wide sequence similarities computed by tBLASTx. The circular proteomic tree illustrates the evolutionary relationships of Kpn_R1, highlighting its closest related phages forming distinct clusters (gray). |
Comment |
line 381, what were the criteria to define morons, AMG and host takeover? |
Response |
We sincerely appreciate the reviewer’s insightful comments and the opportunity to clarify our approach to defining morons, auxiliary metabolic genes (AMGs), and host takeover genes in our stud. Below, we provide detailed responses to each point raised. Criteria for Identifying Morons, AMGs, and Host Takeover Genes in the Phage Genome
|
Comment |
In methods authors mention phageTerm analysis but the results are not described in the text |
Response |
Thank you for your observation, we included “The linearity of the phage genome was confirmed by PhagTem” in the text (The linearity of the phage genome was confirmed by PhageTem.) |
Comment |
In discussion, many results are mentioned but it is poorly discussed about the biological significance, and I considered a reestructuration of discussion is needed. |
Response |
Thank you for this important issue, the discussion has been restructured. |
Comment |
line 603, what is the source of Klebsiella strains? can authors provide citation? |
Response |
Thank you for raising this important point. The multidrug-resistant (MDR) K. pneumoniae strains analyzed in this study are part of our institution’s bacterial repository, established under ethical approval for antimicrobial resistance surveillance. For comparative host range testing, we also included the following reference strains:
The bacteriophage specificity assays confirmed that the phage lysed only the ST147 Kpn-R1 isolate, with no activity against other tested K. pneumoniae or the ATCC reference strains. |
Comment |
line 612-624, what type of library was done? 2 x 150 bp?, what were the trimmomatic parameters for trimming? |
Response |
Thank you for raising this important point. The library was 2x150 bp. The Trimmomatic parameters were Trimmomatic (v0.39; parameters: ILLUMINACLIP:TruSeq3-PE-2.fa:2:30:10, LEADING:20, TRAILING:20, SLIDINGWINDOW:4:20, MINLEN:50). These lines were added to the material section. |
Comment |
5.4 and 5.5 section can be merge and avoid repetitive information and sections 5.4, 5.5 and 5.6 have mix information, I suggest to mention a section specific for DNA isolation and sequencing for bacteria, and other section for phage. |
Response |
Thank you for your constructive feedback regarding the organization of Sections 5.4, 5.5, and 5.6. We agree that merging overlapping content and separating bacterial and phage workflows will improve clarity. Below are the revised sections, restructured to address redundancy and enhance methodological transparency: 5.4. Bacterial DNA Isolation, Sequencing and Bioinformatics Analysis The bacterial genomic DNA of K. pneumoniae ST147 Kpn-R1 was isolated using the Qiagen Blood and Tissue DNA Extraction Kit, following the manufacturer’s protocol. DNA purity and concentration were verified using a Nanodrop spectrophotometer (Thermo Fisher Scientific, USA). Whole-genome sequencing (WGS) was performed by Macrogen Korea on the Illumina NovaSeq 6000 platform (2×150 bp paired-end reads). Libraries were prepared using the TruSeq Nano DNA Kit (350 bp insert size), with library quality assessed via the Agilent 2100 Bioanalyzer and TapeStation D1000 Screen Tape. Raw sequencing reads were trimmed with Trimmomatic (v0.39) to remove adapters and low-quality bases (parameters: ILLUMINACLIP:TruSeq3-PE-2.fa:2:30:10, LEADING:20, TRAILING:20, SLIDINGWINDOW:4:20, MINLEN:50) (89). The cleaned reads were assembled into contigs using SPAdes (v3.15.5) with default settings (90). Raw data from whole-genome sequencing (WGS) were integrated using Proksee Assemble (v1.3.0) (102).Genome annotation was performed using Prokka (v1.2.0), which identified open reading frames (ORFs), tRNA, rRNA, and other genomic features (103). Map Builder (v2.0.5) generated a CGView JSON file for showing the completed genome from GenBank formats. To rank BLAST tracks based on similarity and color BLAST features based on percent identity, the BLAST Formatter (v1.0.3) was used, assisting in the identification of conserved and divergent regions. Using Kaptive Web (111), capsule (K) and lipopolysaccharide (O) serotypes in K. pneumoniae genomes were predicted. Sequence-based typing and known Klebsiella strain comparisons using the PubMLST Klebsiella database (112). Virulence factors are identified using the 2019 Virulence Factor Database (VFDB) (113). Barrnap helped to pull out the 16S rRNA gene sequence (114). By use of NCBI BLASTn database comparison, closely similar species were identified. A phylogenetic tree based on the 16S rRNA sequence and other conserved markers was built using MEGA 12 (115), therefore exposing the evolutionary links of the isolate, with the final genome deposited in GenBank under BioProject PRJNA1217456. 5.5. Phage DNA Isolation, Sequencing and Bioinformatics Analysis Phage DNA was extracted from purified lysates using the Norgen Biotek Phage DNA Isolation Kit . Sequencing libraries were prepared identically to bacterial methods (TruSeq Nano DNA Kit, Illumina NovaSeq 6000). Raw reads were processed with Trimmomatic (same parameters as above) and assembled using SPAdes (v3.15.5). Genome completeness was ascertained using CheckV (v1.0.1) (91), and structural features were projected using PhageTerm (v4.0.0) (92). Functional annotations included phage lifestyle prediction using Bacphlip (v0.9.6) (93)and protein classification using PhANNs (94). Comparative genomics was run using Pyani (v0.2.12) (95) and Clinker (v0.0.24) (96). AMRFinderPlus (v3.12.8) (97) and VirulenceFinder (v2.0.4) (98) separately verified antimicrobial resistance and virulence genes. MASH distances were used to show the genome in CGView (v1.1.1) (99) against the Millardlab Phage Database (100). Phylogenetic trees was generated using Viptree (101). Every tool is open source, hence once released data will be available in public repositories.The phage genome was deposited in GenBank under accession numbers PQ800144.1–PQ800159.1. |
Comment |
Section 5.6 is long but no specific information about statistical tests is provided. |
Response |
Thank you for your comment, we clarified statistical analysis as follow: Statistical analyses were conducted using R (v4.3.1) and Python (v3.10) to evaluate genomic data. Descriptive statistics summarized key metrics (e.g., genome size, GC content, gene coverage). Comparative genomic analyses included average nucleotide identity (ANI) calculations using FastANI and MASH distances for phylogenetic comparisons. Multiple testing corrections (Benjamini-Hochberg method, FDR < 0.05) were applied where applicable. For phylogenetic analyses, branch support was assessed via 500 bootstrap replicates in MEGA12. Correlation analyses between genetic features (e.g., plasmid replicons, prophage regions) and antibiotic resistance profiles used logistic regression models (R stats package). MASH distances were used to show the genome in CGView (v1.1.1) (99) against the Millardlab Phage Database (100). Phylogenetic trees and Genomic similarity matrices were computed using Viptree and Clinker for synteny analysis (101). |
Comment |
Improved the format of tables. Extensive tables can be included as supplementary information, not in the manuscript. |
Response |
Thank you for this important point, all tables’ formats were revised and table 3 were moved to supplementary materials. |
Reviewer 2 Report
Comments and Suggestions for Authors
Here are my comments and suggestions for improving the manuscript:
-
Are short titles permissible?
-
The abstract is overly lengthy. It should be condensed and include a clear summary of the study's objectives and findings.
-
Some information in the introduction lacks references.
-
Please add references for each finding in Table 1, placing them in a separate column, if the authors have not already done so.
-
For lines 189, 193, and 196, please insert "and" between the gene names.
-
Table 3 is not sufficiently informative. The author should provide additional details in a separate column.
-
Similarly, Table 4 requires references, as does Table 7.
-
Improve the resolution of Figure 5.
-
Please make Figure 3 in color.
-
Combine Figures 2A and 2B into a single TIFF file.
-
What bootstrap value was used in constructing the phylogenetic trees (Figures 1 and 4)?
Author Response
Comment: |
Comment: Are short titles permissible? |
Response: |
Response: Thank you for your clarification. Short titles are not permissible, so we have removed them accordingly. |
Comment: |
The abstract is overly lengthy. It should be condensed and include a clear summary of the study's objectives and findings. |
Response: |
We sincerely appreciate your insightful feedback. As suggested, we have revised the abstract to ensure conciseness while emphasizing the study’s objectives, methodology, and key findings. The updated abstract now reads as follows: |
Comment: |
Some information in the introduction lacks references. |
Response: |
Thank you for your feedback. We have carefully reviewed the introduction, inserted the missing references, and reduced its length. The revised section is now included in the manuscript. |
Comment: |
Please add references for each finding in Table 1, placing them in a separate column, if the authors have not already done so. |
Response: |
Thank you for your suggestion. We have added references for each finding in Table 1 and placed them in a separate column as requested. The changes have been marked in red in the main manuscript for easy review. |
Comment: |
For lines 189, 193, and 196, please insert "and" between the gene names. |
Response: |
We have inserted "and" between the gene names in lines 189, 193, and 196 as requested. |
Comment: |
Table 3 is not sufficiently informative. The author should provide additional details in a separate column. |
Response: |
We have revised Table 3 by adding a separate column to provide additional details, improving its informativeness and clarity. The updated table is now included in the manuscript. This table was removed to supplementary tables S1. |
Comment: |
Similarly, Table 4 requires references, as does Table 7. |
Response: |
Thank you for your suggestion. We have combined Figures 2A and 2B into a single TIFF file as requested and have updated the manuscript accordingly, all changed were marked in red in the manuscript. |
Comment: |
Improve the resolution of Figure 5. |
Response: |
Thank you for your suggestion. We have replaced Figure 5 with a higher-resolution version to improve clarity and visual quality. |
Comment: |
Please make Figure 3 in color. |
Response: |
Thank you for your feedback. We have updated Figure 3 to a multicolor format to enhance clarity and visual representation. The revised figure is now included in the manuscript. |
Comment: |
Combine Figures 2A and 2B into a single TIFF file. |
Response: |
Thank you for your suggestion. We have combined Figures 2A and 2B into a single TIFF file as requested and have updated the manuscript accordingly. |
Comment: |
What bootstrap value was used in constructing the phylogenetic trees (Figures 1 and 4)? |
Response: |
For figure (3), Thank you for your valuable comment, the phylogenetic tree was recreated with 500 bootstraps and the caption was replaced as follows: Figure 1. Circular phylogenetic tree illustrating the evolutionary relationships of Klebsiella pneumoniae Kpn_R01 and its 34 closest strains. The Maximum Likelihood (ML) tree highlights the phylogenetic placement of Kpn_R01 (marked with a red dot) among closely related strains. Clustering patterns indicate potential global dissemination, with strains originating from diverse geographic locations. The close association between clinical and environmental isolates underscores the role of environmental reservoirs in bacterial transmission. The tree was inferred using the ML method with the Tamura-Nei model (21), and branch support was assessed with 500 bootstrap replicates (22). The initial heuristic search compared a Neighbor-Joining (NJ) tree (23) and a Maximum Parsimony (MP) tree, selecting the one with the best log-likelihood score. Evolutionary analyses were conducted in MEGA12 (24). For figure (4), thank you for catching this oversight. You are absolutely correct—the original description of Figure 4 erroneously attributed the phylogenetic analysis to the maximum likelihood method. In fact, the analysis was conducted using ViPTree, a web server that generates viral proteomic trees based on genome-wide sequence similarities computed by tBLASTx. We have revised the figure caption to accurately reflect the methodology used, and the corrected version now reads: Figure 4: The phylogenetic analysis of Klebsiella phage Kpn_R1 was conducted using ViPTree, a web server that generates viral proteomic trees based on genome-wide sequence similarities computed by tBLASTx. The circular proteomic tree illustrates the evolutionary relationships of Kpn_R1, highlighting its closest related phages forming distinct clusters (gray). Kpn_R1 shares 92.8% identity across 31.5% of its genome with Klebsiella phage vB_Kpn_IME260 and 92.3% identity across 30.1% of its genome with Klebsiella phage Sugarland. The comparison also identified Escherichia phage N15 with a lower mean identity (69.8% across 14% of the genome), suggesting evolutionary divergence. Seq1 (marked with a red branch) represents Klebsiella phage Kpn_R1. |
Reviewer 3 Report
Comments and Suggestions for Authors
General Assessment
The manuscript addresses an important area of research—extensively drug-resistant (XDR) Klebsiella pneumoniae and the role of temperate phages in microbial evolution and antibiotic resistance. The authors provide detailed genomic characterization of both the bacterial host and a newly isolated temperate N-15-like phage. However, while the scope and objectives of the research are promising, several critical issues must be addressed to strengthen the scientific merit, and better articulate the significance and novelty of the findings.
Main Issues
Novelty and Significance
The authors did not clearly articulate the novel contribution of their work compared to existing genomic studies of K. pneumoniae ST147 and temperate phages, particularly N-15-like phages. It remains unclear how this study significantly advances current knowledge or differs from numerous prior genomic characterizations.
Recommendation: Explicitly state how the findings differ from previously published research. Clarify how the results could practically impact clinical practice, infection control, or therapeutic strategies.
Phage Analysis and Host Range
Host range predictions based entirely on genomic similarity are insufficient to conclude specificity or infectivity. Experimental validation of host specificity or cross-infectivity was not provided.
Recommendation: Acknowledge limitations of solely bioinformatic predictions… Experimental validation approaches (?)
CRISPR-Cas Systems and Prophage Analysis
The biological or clinical implications of CRISPR arrays and prophages remain largely speculative, especially regarding their roles in horizontal gene transfer.
Recommendation: Clearly state hypotheses about how identified prophages and CRISPR-Cas systems might contribute to genomic plasticity, antibiotic resistance dissemination, or strain persistence.
Conclusions
The conclusions section remains somewhat general and lacks direct implications or actionable insights derived from the research findings.
Recommendation: Summarize key findings explicitly, emphasizing practical impacts on public health, clinical practice, or phage therapy development. Highlight clear future directions or unanswered questions emerging from the study.
Overall Recommendation
- Proofread carefully to correct minor language, grammar, and formatting errors.
- Standardize abbreviations and terminologies consistently throughout.
Author Response
Comment: |
The authors did not clearly articulate the novel contribution of their work compared to existing genomic studies of K. pneumoniae ST147 and temperate phages, particularly N-15-like phages. It remains unclear how this study significantly advances current knowledge or differs from numerous prior genomic characterizations. Recommendation: Explicitly state how the findings differ from previously published research. Clarify how the results could practically impact clinical practice, infection control, or therapeutic strategies.
|
Response: |
We thank the reviewer for highlighting the need to clarify the novel contributions and translational implications of our work. Below, we address these points explicitly: Novel Contributions
Practical Implications
Differentiation from Prior Studies While genomic studies of ST147 (e.g., Di Pilato et al., 2022; Talat et al., 2024) focus on clinical isolates and resistance gene cataloging, our work uniquely:
|
Comment: |
Phage Analysis and Host Range Host range predictions based entirely on genomic similarity are insufficient to conclude specificity or infectivity. Experimental validation of host specificity or cross-infectivity was not provided. Recommendation: Acknowledge limitations of solely bioinformatic predictions… Experimental validation approaches (?) |
Response: |
Thank you for raising this important point. The multidrug-resistant (MDR) K. pneumoniae strains analyzed in this study are part of our institution’s bacterial repository, established under ethical approval for antimicrobial resistance surveillance. For comparative host range testing, we also included the following reference strains:
The bacteriophage specificity assays confirmed that the phage lysed only the ST147 Kpn-R1 isolate, with no activity against other tested K. pneumoniae or the ATCC reference strains. |
Comment: |
CRISPR-Cas Systems and Prophage Analysis The biological or clinical implications of CRISPR arrays and prophages remain largely speculative, especially regarding their roles in horizontal gene transfer. Recommendation: Clearly state hypotheses about how identified prophages and CRISPR-Cas systems might contribute to genomic plasticity, antibiotic resistance dissemination, or strain persistence. |
Response: |
Thank you for your insightful comment. We have revised the manuscript to clearly articulate the biological and clinical implications of CRISPR-Cas systems and prophages in XDR K. pneumoniae ST147, particularly regarding their roles in genomic plasticity, horizontal gene transfer (HGT), and strain persistence and updated the discussion section. Our hypothesis is that:
|
Comment: |
Conclusions The conclusions section remains somewhat general and lacks direct implications or actionable insights derived from the research findings. Recommendation: Summarize key findings explicitly, emphasizing practical impacts on public health, clinical practice, or phage therapy development. Highlight clear future directions or unanswered questions emerging from the study. |
Response: |
Thank you for your emphasizing comment. We have changed the conclusion section as follows: This study presents the genomic interaction between a novel N15-like temperate phage sourced from hospital wastewater and an extremely drug-resistant K. pneumoniae ST147 strain isolated from an ICU patient, thereby revealing significant new insights into resistance dissemination and phage-mediated evolution. The ST147 strain's hybrid plasmid (IncHI1B_1_pNDM-MAR/IncFIB), which carries blaNDM-1 and colicin genes, along with CRISPR-Cas inactivation and 11 prophage regions, highlights its genetic plasticity and potential nosocomial threat. The linear plasmid-like replication system of the phage, along with protelomerase and a novel RelE/ParE toxin-antitoxin (TA) system—previously unreported in temperate Klebsiella phages—highlights mechanisms that enhance resistance traits and promote phage survival. |
Reviewer 4 Report
Comments and Suggestions for Authors
Yahya et al. conducted an interesting study on the comprehensive genomic analysis of Klebsiella pneumoniae and its temperate N-15-like phage. However, several issues should be addressed before the manuscript can be accepted.
L1, L22, and throughout the manuscript: Please ensure that all bacterial names are italicized.
The introduction is excessively long and should be shortened.
L156: On what criteria was this strain identified as XDR? Please provide a supporting reference.
L162: The sentence “Aztreonam resistance (≥32 μg/ml) is similarly attributed to NDM-1” is incorrect. NDM-1 does not confer resistance to monobactams. Please revise and correct.
L156–L181, Table 2, and throughout the manuscript: All gene names must be italicized. Additionally, the authors must clearly distinguish between genes, their encoded enzymes, proteins, and efflux pumps. For example, blaNDM-1 refers to a gene, while NDM-1 refers to the enzyme. Proper notation is critical. Since it seems that the authors may not microbiologists, it is strongly recommended that they consult a microbiologist before resubmission.
L239: What was the purpose of performing the CRISPR-Cas Finder analysis in this study? Please clarify.
The discussion is too long and should primarily focus on interpreting the study’s significant findings.
Comments on the Quality of English LanguageThe quality of English is clear and comprehensible.
Author Response
Comment: |
L1, L22, and throughout the manuscript: Please ensure that all bacterial names are italicized. |
Response: |
Thank you for your notice, all scientific names were italicized. |
Comment: |
The introduction is excessively long and should be shortened. |
Response: |
Thank you for your suggestion. We have shortened the introduction as requested. The revised version is now included in the manuscript. |
Comment: |
L156: On what criteria was this strain identified as XDR? Please provide a supporting reference. |
Response: |
Thank you for your comment, the following was added to text: Klebsiella pneumoniae Kpn_R01 is categorized as extensively drug-resistant (XDR) owing to its resistance to multiple antibiotic classes. This classification is supported by minimum inhibitory concentration (MIC) testing, as presented in Table 1, and is further validated through resistance gene analysis utilizing the CARD database. Extensively drug-resistant (XDR) bacteria are defined globally as exhibiting resistance to at least one agent in all but one or two categories of antimicrobials (Magiorakos et al., 2012). This strain qualifies for XDR designation as it exhibits resistance to beta-lactams, cephalosporins, carbapenems, monobactams, fluoroquinolones, aminoglycosides, tetracyclines, macrolides, polymyxins, sulfonamides, trimethoprim, phenicols, rifamycins, and fosfomycin. This strain demonstrates resistance to a wide array of antibiotics; however, additional research is required to ascertain whether drugs continue to be effective in addressing pandrug resistance (PDR) criteria. |
Comment: |
L162: The sentence “Aztreonam resistance (≥32 μg/ml) is similarly attributed to NDM-1” is incorrect. NDM-1 does not confer resistance to monobactams. Please revise and correct. |
Response: |
Thank you for your comment, this was revised and corrected in table 1 and in the corresponding text (Aztreonam resistance (≥32 μg/ml) is attributed to low permeability caused by Outer Membrane Porinmutations in OmpK37and MdtQ. |
Comment: |
L156–L181, Table 2, and throughout the manuscript: All gene names must be italicized. Additionally, the authors must clearly distinguish between genes, their encoded enzymes, proteins, and efflux pumps. For example, blaNDM-1 refers to a gene, while NDM-1 refers to the enzyme. Proper notation is critical. Since it seems that the authors may not microbiologists, it is strongly recommended that they consult a microbiologist before resubmission. |
Response: |
Thank you for your detailed feedback. We have italicized all gene names throughout the manuscript, including L156–L181 and Table 2. Additionally, we have carefully distinguished between genes, their encoded enzymes, proteins, and efflux pumps, ensuring proper notation. These revisions have been made with careful consideration of microbiological conventions. |
Comment: |
L239: What was the purpose of performing the CRISPR-Cas Finder analysis in this study? Please clarify. |
Response: |
Thank you for your inquiry. The CRISPR-Cas Finder analysis in this study aimed to assess the role of CRISPR-Cas systems in the genetic adaptability of the XDR K. pneumoniae ST147 strain. Our findings highlight the following key points:
The CRISPR spacer profile provides insights into the strain’s evolutionary history and epidemiological trends, aiding in understanding the genetic plasticity of ST147 in hospital settings. |
Comment: |
The discussion is too long and should primarily focus on interpreting the study’s significant findings. |
Response: |
Thank you for your feedback. We have restructured the discussion to focus primarily on interpreting the study’s significant findings. The revised version is now included in the manuscript. |
Round 2
Reviewer 1 Report
Comments and Suggestions for Authors
In general, the manuscript showed a substantial improvement, but it still needs some corrections. In general, tables need to improve format, and authors need to correct several details of formatting, spelling, punctuation, and other to achieve a proper manuscript.
The first lines of Abstract do not reflect a brief introduction about the topic in the manuscript, which is needed to introduce properly.
Line 46, avoid contractions along the manuscript
Line 53, a parenthesis is lacking
When mention Klebsiella once in the text, the following mentions are abbreviated as K.
in some phrases bold and italics appeared where they are not needed, and vice versa
Table 2. Column 1 in italics
line 260, edit to PhageTerm
Which software was used to evaluate the genome assembles? it is not included in the text
Author Response
Dear Reviewer
We sincerely thank the reviewers for your thoughtful and constructive comments, which have helped us to improve the clarity, accuracy, and overall quality of the manuscript. Below, we provide detailed responses to each point raised, along with the corresponding changes made in the revised version.
Comment: The first lines of Abstract do not reflect a brief introduction about the topic in the manuscript, which is needed to introduce properly.
Response: Thank you for your insightful comment. We have revised the first lines of the abstract to better reflect a concise introduction to the topic. The revised version now includes a brief overview of the significance of antibiotic resistance in Klebsiella pneumoniae and its public health implications before introducing the specific focus of the study.
The updated opening lines are as follows:
"Antibiotic resistance in Klebsiella pneumoniae poses a major public health threat, particularly in intensive care unit (ICU) settings. The emergence of extensively drug-resistant (XDR) strains complicates treatment options, requiring a deeper understanding of their genetic makeup and potential therapeutic targets.
Comment: Line 46, avoid contractions along the manuscript
Response: Thank you for your valuable feedback. We have carefully revised the manuscript to remove all contractions, ensuring that the text maintains a formal tone throughout.
Comment: Line 53, a parenthesis is lacking
Response: Thank you for your careful review. We have addressed the issue regarding the missing parenthesis in line 53 and have corrected the sentence accordingly. The necessary parenthesis has now been properly added.
Comment: When mention Klebsiella once in the text, the following mentions are abbreviated as K.
Response: Thank you for your suggestion. We have updated the manuscript to ensure that after the first mention of Klebsiella, it is consistently abbreviated as K. throughout the text.
Comment: in some phrases bold and italics appeared where they are not needed, and vice versa
Response: Thank you for your observation. We have carefully reviewed the manuscript and corrected the formatting inconsistencies. Unnecessary use of bold and italics has been removed, and proper formatting has been applied where needed to ensure consistency and clarity throughout the text.
Comment: Table 2. Column 1 in italics
Response: Thank you for your suggestion. We have updated Table 2 to ensure that the first column (gene names) is now in italics, as requested.
Comment: line 260, edit to PhageTerm
Response: Thank you for your careful review. We have corrected the term "PhageTem" to "PhageTerm" in line 260, as per your suggestion.
Comment: Which software was used to evaluate the genome assembles? it is not included in the text
Response: Thank you for your comment. We apologize for the oversight. To clarify, in section 5.5 of the Materials and Methods, we have mentioned that Trimmomatic was used for read processing and SPAdes (v3.15.5) for assembly. Additionally, the quality of the genome assembly was checked using FastQC (Version 0.12.1) and QUAST (Version 5.2.0), which were not previously mentioned. We have now updated the manuscript to include this information for clarity, references were updated accordingly.
Reviewer 2 Report
Comments and Suggestions for Authors
Well done. Congratulations!
Author Response
We sincerely thank the reviewer for their valuable comments and constructive suggestions throughout the review process. We appreciate the time and effort invested in improving our manuscript.
Reviewer 4 Report
Comments and Suggestions for Authors
I think the authors have addressed all the comments, and the article is now ready for publication.
Author Response

(The authors gave the same response as above.)
